# Urban Distribution and Evolution of the Yangtze River Economic Belt from the Perspectives of Urban Area and Night-Time Light

Huimin Xu [1,2,*], Shougeng Hu [2] and Xi Li [3]

1   School of Economics, Wuhan Polytechnic University, Wuhan 430048, China
2   School of Public Administration, China University of Geosciences, Wuhan 430074, China
3   State Key Laboratory of Information Engineering in Surveying, Mapping and Remote Sensing, Wuhan University, Wuhan 430079, China
*   Correspondence: shannon_x@whpu.edu.cn

**Abstract:** Research on urban development patterns and urban sprawl in the Yangtze River Economic Belt (YREB) has received wide attention. However, existing research has always made use of statistical data, which are not often available. Considering the high availability of satellite data, this study attempts to combine two satellite-acquired indexes, including urban area and night-time light, to evaluate the urban development of the YREB during 2012–2019. The methods included using growth index, rank-size law, and the Markov transition matrix, as well as constructing urban night-time light density and unbalanced index of night-time light, derived from the Gini Index. Some important patterns were revealed. Firstly, the three reaches (Upper Reaches, Middle Reaches, and Lower Reaches) in the YREB have all shown rapid growth in urban area and night-time light, and they all have increased in urban density. Secondly, from the perspective of regional disparity, the Upper Reaches have the highest growth rate of the urban area, while the Middle Reaches have the highest growth rate of night-time light; and the Upper Reaches have more urban sprawl, while the Middle Reaches have shown more compact growth. Thirdly, higher urban density is related to more balanced development across cities. Our study suggests new knowledge can be obtained by combining the two indexes for understanding urban development in the YREB.

**Keywords:** Yangtze River Economic Belt; urban development; urban area; night-time light; VIIRS

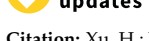



## 1. Introduction

Since its Reform and Opening-up in 1978, China has experienced fast economic growth along with urbanization over the last four decades. Urbanization has increasingly become the main engine of urban economic development in China [1]. Therefore, the expansion of urban land is playing an essential role in economic growth at multi-scales. However, the rapid urbanization in China has resulted in accelerating urban sprawl, which is threatening the environment, the ecosystem, and socioeconomic sustainability. Generally, urban sprawl refers to the development of low density and low efficiency urban expansion, leading to inefficient land resource utilization [2]. Urban sprawl causes a series of environmental and social problems, such as traffic congestion [3], excess carbon emission [4,5], and environmental pollution [2,3,6]. The studies on urban sprawl have attracted increased attention. For example, Fulton et al. proposed an urban sprawl index (USI) by comparing the matching degree between the growth rates of the urban population and the urban area [7], which has been widely used in different regions to measure urban sprawl [8–10]. Li et al. integrated urban land census data and urban population data to investigate the pattern of urban sprawl and find disparities [11]. As urban sprawl refers to urban expansion with low efficiency, urban efficiency has been widely studied for urban development. From the perspective of land use supply, high urban efficiency is reached when a given

output, such as GDP, is achieved with minimum input from urban land resources, or when maximum output is produced based on given input. To evaluate urban efficiency, the most frequently used methods are data envelopment analysis (DEA), including the traditional DEA model [12], the super-efficiency DEA model [13], and the slacks-based measure (SBM) model [14], etc.

The Yangtze River Economic Belt (YREB) is defined as the region comprising the economic areas around the Yangtze River, which is the longest river in China [15], and the YREB was promoted in priority of state strategy in 2014. The YREB covers 11 provincial regions with an area of 2.05 million km$^2$. At present, studies on the YREB mainly include urbanization [16–18], ecological environment [19–22], and sustainable development [23–25]. Urbanization in the Yangtze River Economic Belt has attracted widespread attention in the academic community and government. In terms of the relationship between urban expansion and the economy, Xie et al. explored the impact of urban construction land growth on regional economic growth in the YREB using spatial analysis and the econometric model, and the results suggest that urban expansion in the YREB has had a positive impact on economic growth [16]. Liu et al. explored the relationship between urban land expansion and the scope of human activities in the YREB based on Landsat and nocturnal light data [26]. As for urban spatial patterns, Li et al. mapped their evolution in the YREB using the rank-size law and the unbalanced index [27]. Guan et al. simulated and predicted the urban sprawl trend in the YREB [15], suggesting that the urban area in the YREB will continue to spread and present unbalanced patterns.

Under continuous urban sprawl, an in-depth analysis of the urban expansion and efficiency in the urban development in the YREB is of great practical significance. In the past stage of rapid urbanization growth, the extensive expansion of cities in the YREB has led to problems such as the waste of urban land resources, which impeded the improvement of urban efficiency [28]. Some recent research has used a variety of methods to assess urbanization efficiency in the YREB. For example, Jin et al. applied stochastic frontier analysis to evaluate the urbanization efficiency of more than 100 cities over the YREB [17], and it suggests that the urban efficiency shows a trend toward growth, but it also shows large intra-provincial and inter-provincial variations. Liu et al. adopted the super efficiency SBM model to analyze urban land use efficiency in 11 provinces and cities in the YREB [28]. Wang et al. used the space-time interaction method to explore the match between land expansion and population growth in the entire YREB and the major urban agglomerations [29].

Evaluating urban efficiency of the YREB is important for formulating reasonable plans for sustainable urban development in the region. Therefore, establishing a technical framework for evaluating the urbanization process in the YREB is urgently needed. However, previous studies measuring urban expansion and land-use efficiency in the YREB, made use of statistical data, such as investment data, GDP and population, which are not always available for some regions and years in YREB. On contrast, remote sensing data from satellite is acquirable at large scale across different years, among which the satellite-observed night-time light can be viewed as output of urban development [30,31] and thus it can be utilized to measure the urban efficiency. This study attempts to measure the urban growth patterns and urban efficiency in YREB from two different satellite products, night-time light imagery and impervious maps derived from Landsat images. These datasets are completely derived from satellite observation, making them available and comparable in both spatial and temporal dimensions. Based on these data sets, we will analyze the spatiotemporal patterns of urbanization in YREB during 2012 and 2019 and evaluate urban growth efficiency by constructing a new index. This methodology may help to provide a new way to measure urban growth and urban efficiency in YREB as well as other regions.

## 2. Materials and Methods

### 2.1. Study Area

The concept of YREB was first proposed in 2013 and was promoted to state strategy in 2014. The Yangtze River Economic Belt is also the largest economic zone in China with the highest economic density. It is responsible for realizing China's future economic sustainable development. The YREB, as illustrated in Figure 1, includes nine provinces (e.g., Jiangsu, Zhejiang, Anhui, Jiangxi, Hubei, Hunan, Sichuan, Yunnan, and Guizhou) and two municipals (e.g. Chongqing and Shanghai). The YREB is divided into three sub-regions, Upper Reaches (Sichuan, Yunnan, and Guizhou and Chongqing), Middle Reaches (Jiangxi, Hubei and Hunan) and Lower Reaches (Jiangsu, Zhejiang, Anhui and Shanghai). Among them, the Upper Reaches includes Yangtze River Delta Urban Agglomeration, which is the largest urban agglomeration with strong economic vitality and the highest developed region in China.

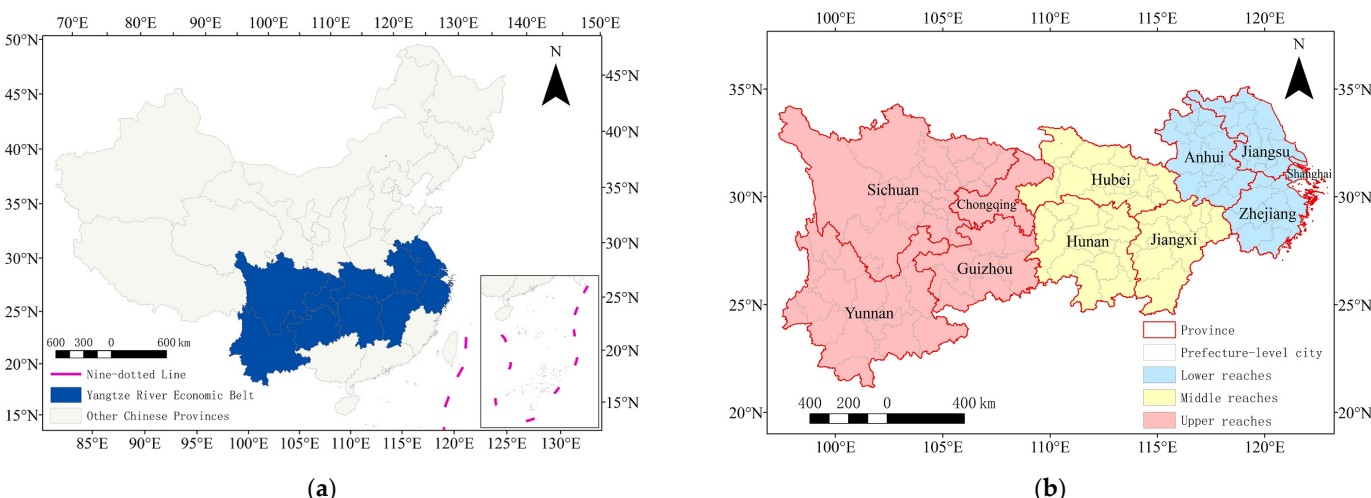

(**a**)　　　　　　　　　　　　　　　　　(**b**)

**Figure 1.** The map of the Yangtze River Economic Belt: (**a**) the location of YREB in China; (**b**) the administrative regions of YREB.

### 2.2. Study Data

In this study, there are two main data sources. The night-time light images and impervious surface map.

#### 2.2.1. Night-Time Light Images

Previous studies show that night-time light in cities are highly correlated to GDP, electricity consumption, and urban population [32,33]; thus it is viewed as a comprehensive proxy for urban and economic development. Furthermore, considering that the statistical data have some bias and are sometimes not comparable in temporal and spatial dimension, night-time light has widely been used as a proxy for urban and social development [34–37]. Accordingly, night-time light images have been increasingly used in the urbanization research of the YREB. For example, Xu et al. evaluated the urbanization process of the YREB at different scales using DMSP/OLS night-time light images [38]. Zhong et al. adopted the landscape index, standard deviation ellipse and spatial correlation analysis to quantify the spatial and temporal evolution of urban land expansion and its driving factors in the YREB by using DMSP/OLS night-time light images [39]. Our study also employed night-time light as the index to measure the economic activities in the urban area of YREB.

We select the Black Marble product suite, produced by National Aeronautics and Space Administration (NASA), as the night-time light images, which have been widely used in recent years [40,41]. The Black Marble is produced from Day/Night band (DNB) of Visible Infrared Imaging Radiometer (VIIRS). The annual composites (VNP46A4 product) of the Black Marble was produced by averaging the daily product (VNP46A2) in which stray

light, moonlight, atmospheric effects and vegetation effect were removed [42]. Normally, clouds will impact quality of the daily product because the satellite is not able to receive a night-time light signal in cloudy weather, but this effect is not significant in the annual composite (VNP46A4), as it is rare that a region is covered by clouds through all the days in a year; therefore, the annual composites are robust to reflect the spatio-temporal distribution of night-time light. The All-Angle_Composite_Snow-Free product from the VNP46A4 was selected, which is downloadable at the NASA website (https://ladsweb.modaps.eosdis.nasa.gov/, accessed on 27 December 2022). In the Black Marble product, the global areas were divided into image tiles, with each tile covering 10° × 10° area. To cover the entire YREB each year, eight tiles were downloaded, and the images for 2012 and 2019 were mosaicked as shown in Figure 2.

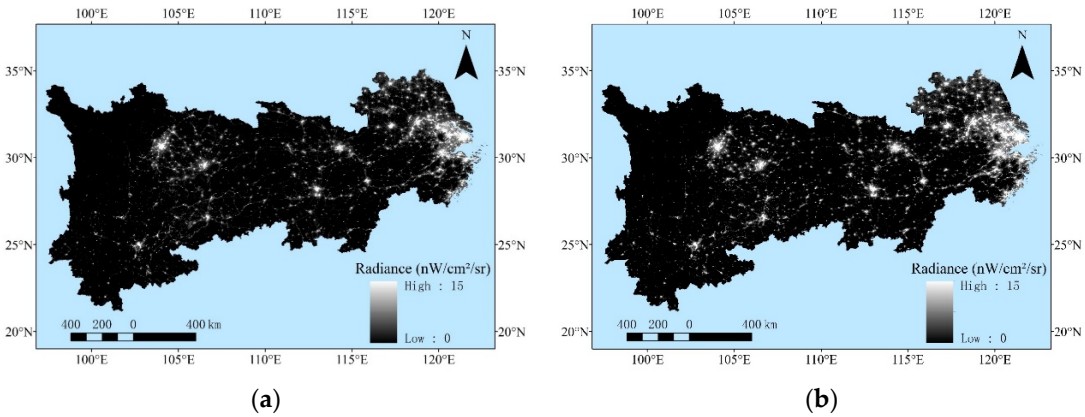

**Figure 2.** The night-time light images for YREB: (**a**) 2012; (**b**) 2019. Note: the radiance higher than 15 was set to 15 for demonstration.

### 2.2.2. Impervious Surface Data

In urban expansion and urban efficiency analysis, the urban area is viewed as a major index for urban development. Although statistical books on China provide urban area data, their comparability is not reliable because the statistical scale varies among different cities.s To overcome this problem, we use a satellite-derived impervious map in this study, as satellite remote sensing is able to map the urban extent accurately without human interferences. The data set is 30-m time-series globally impervious surface area (GISA) [43] and can be downloaded at Zenodo (https://zenodo.org/record/6476661#.Y1Sh9bZBw2y, accessed on 27 December 2022). The dataset, derived from Landsat images at 30 m resolution, was validated with accuracy of 93% [43], suggesting that it is feasible for evaluating urban expansion. The impervious maps for 2012 and 2019 were shown in Figure 3. We will transform the impervious map to urban land cover data in the following section.

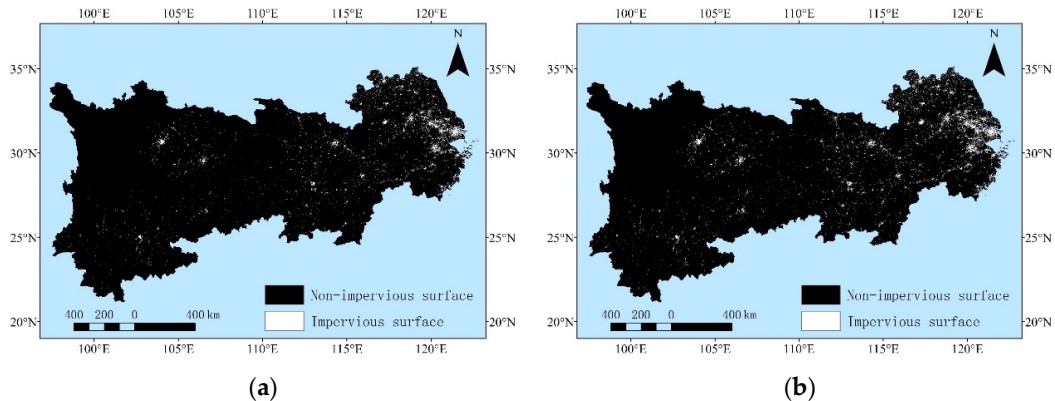

**Figure 3.** The impervious surface maps for YREB: (**a**) 2012; (**b**) 2019.

*2.3. Methodology*

2.3.1. Data Preprocessing

For the Black Marble, the product of night-time light used in this study, the original data format is HDF5, and it is converted to the GeoTiff format for convenience in analysis. The mosaicked images were reprojected into Albers Equal Area Projection in WGS84 coordinates, and the spatial resolution is 500 m.

For the GISA dataset, it is composed of tiles with $5° \times 5°$ spatial coverage. The original spatial resolution is 1 arc second. The downloaded tiles were mosaicked; they were reprojected onto the Albers Equal Area projection in WGS 84 coordinates, and the final spatial resolution was 30 m. Consequently, we converted the binary impervious map (value in 0 or 1) at 30 m resolution into a proportion map of impervious surface at 500 m resolution. Finally, an urban mask is defined as pixels, where proportion of impervious surface is larger than 20%, in the National Land Cover Dataset by the United States Geological Survey [44]; it has been successfully applied to mapping urban extent on a large scale [45]. The urban mask, at 500 m resolution, will be used to calculate urban areas in the following sections.

2.3.2. Measuring Urban Size and Density from Night-Time Light

In this study, we used two satellite-acquired indexes to measure the urban size with two indexes, urban area and total urban night-time light. The urban area is calculated from the urban mask of a city as defined in Section 2.3.1. The total night-time light is calculated as

$$\text{TNL} = \sum_{i=1}^{n} rad_i \tag{1}$$

where TNL denotes the total night-time light inside an urban mask, $rad_i$ denotes radiance of $i$th pixel of the VIIRS image, and $n$ denotes the number of VIIRS pixels inside the urban mask. The night-time light per area for a city is defined as

$$\text{Density} = \frac{\text{TNL}}{\text{Area}} \tag{2}$$

where Area denotes the urban area of a city. Considering that the unit of TNL is $\text{nW}/\text{cm}^2/\text{sr}$ and unit of area is $\text{km}^2$, the unit of Density is $\text{nW}/\text{cm}^2/\text{sr}/\text{km}^2$, which is overly complex to understand. Therefore, we use the number of urban pixels to replace the urban area in Equation (2), by considering that Area = $n\text{S}_{\text{pixel}}$ where $n$ is the number of urban pixels and $\text{S}_{\text{pixel}}$ is the pixel size which is a constant in this analysis. Therefore, we constructed urban night-time light density (UNLD) as:

$$\text{UNLD} = \frac{\text{TNL}}{n} \tag{3}$$

where UNLD denoted the night-time light density inside an urban region, TNL denoted the total night-time light inside an urban area defined by an urban mask, and $n$ denoted the number of pixels inside the urban area. UNLD is proportional to night-time light per urban area as defined in Equation (2), and thus it can represent the light density in the urban area. The advantage of using UNLD to reflect the urban density is that its unit is $\text{nW}/\text{cm}^2/\text{sr}$, which is understandable from a physical perspective. The design of UNLD is to represent urban density and urban efficiency, as a previous study indicates that night-time light per urban area can represent the GDP density [30], of which the higher value represents higher urban efficiency.

2.3.3. Measuring Distribution of Urban Size

In this study, we used two methods, rank-size analysis and the Markov transition matrix to analyze the urban size distribution and its evolution in YREB.

- Rank-size law

The rank-size law has been widely utilized to describe distribution of urban size, especially to quantify the degrees of agglomeration or dispersion of cities in a region. The rank-size law was firstly proposed by Felix Auerbach [46], and was then improved by Singer [47] and Zipf [48]. A general rank-size law is presented using the following equation:

$$P_i = P_1 R_i^{-q} \tag{4}$$

The above equation can be logarithmized to:

$$\lg P_i = \lg P_1 - q \lg R_i \tag{5}$$

where $P_i$ denotes the size of the $i$th largest city, $P_1$ denotes the size of the largest city, $R_i$ denotes the rank of the $i$th largest city, and $q$ is the Zipf index. The Zipf index is important for reflecting urban size distribution, and it is equal to one when urban size is optimally distributed. Big cities are more agglomerated when $q$ is larger than one, and they are dispersed when it is less than one. The Zipf index is the key index showing the rank-size law. Historically, many variables, including population, urban area, GDP and night-time light have been used to reflect urban sizes and thus can be the inputs for $P_i$ [49–52]. In this study, urban area and total night-time light (TNL) are both employed to describe the urban sizes.

- Markov transition matrix

The rank-size analysis is able to show the entire distribution of urban size inside a region, but the dynamics of individual cities and their statistics cannot be reflected in the analysis. Here, we employed the Markov transition matrix to analyze how cities transformed from one level into another. A typical Markov transition matrix is used to analyze the urban size transition from year to the next year (or 2 adjacent years) [53]. Night-time light has some irregular fluctuations during short periods due to sensor degradation or errors in producing image composition [54], resulting in the phenomenon that the night-time light may show a small decline from one year to the next year in cities with economic growth. A simple way to reduce this kind of abnormality is analyzing night-time light change over a long period rather than a short period [35], so that we only focus on urban size transition during 2012–2019, in which night-time light changes in most of the cities are large enough to suppress the data abnormality. In other words, we only compare the night-time light in 2012 and 2019, in which the effect of data abnormality is relatively small compared to the real night-time light change.

To take Markov transition analysis, we sort all cities into types A, B, C and D based on previously established criteria [55,56]:

$$x_i = \left\{ \begin{array}{l} A, s_i \le 0.5m \\ B, 0.5m < s_i \le m \\ C, m < s_i \le 2m \\ D, 2m < s_i \end{array} \right\} \tag{6}$$

where $s$ is the size of $i$th city, $m$ is the average size of all cities, and $x_i \in \{A, B, C, D\}$ defining the city type. For the year 2012, each city has a type based on Equation (6), and the city has a same or different type in the year 2019. In the $i$th column and $j$th row of the transition matrix, the value $n_{ij}$ denotes the number of cities which were classified into Class $i$ in 2012 and into Class $j$ in 2019. In this study, both urban area and total night-time light were used to describe the urban size so that the transition matrices were generated from the two indexes.

### 2.3.4. Measuring Urban Growth Patterns

We calculated the change rates of three variables including urban area, total night-time light, Urban Night-time Light Density (UNLD), and Unbalanced Index of Night-time Light

(UINL), which will be defined in Section 2.3.6, to evaluate the urban growth patterns in the YREB. The growth rate is defined as:

$$r_{i,j} = \frac{x_j - x_i}{x_i} \tag{7}$$

where $r_{i,j}$ denotes growth rate of $x$ in a region between $i$th year and $j$th year, $x_i$ and $x_j$ denotes the value of $x$ in the $i$th year and $j$th year, respectively. Based on the Equation (3), urban area grows slower than the night-time light if the growth rate of UNLD is positive, and urban area grows faster than the night-time light if the growth rate of UNLD is negative. Therefore, the negative growth of UNLD reflects over-growth of urban area compared to the night-time light, indicating that urban sprawl occurs in the city and urban efficiency is decreased. We have to note that $i$ and $j$ are set to 2012 and 2019, respectively, in this study.

2.3.5. Spatial Statistics

- Spatial autocorrelation analysis

In this study, we need to analyze spatial patterns of urban growth, especially the aggregation pattern, by employing different indexes. Spatial autocorrelation analysis can explore the aggregation, discrete, or random distribution of elements according to their location and value [57]. By calculating spatial autocorrelation index, geographic patterns can be described from qualitative to quantitative. Specifically, Global Moran's I and local Moran's I can well represent spatial similarity or spatial disparity, which will be utilized in this study. For evaluating urbanization in the YREB, the spatial autocorrelation analysis will help to explore the aggregation of different indexes and their change rate, which are related to urban forms and growth. Technical details of spatial autocorrelation analysis can be found in the literature [58].

- Geographically weighted regression

Geographically weighted regression (GWR) is a spatial analysis method used to address spatial heterogeneity, which explores the spatial variation and related driving factors by establishing a local regression equation at each geographic location. Since GWR takes into account the local effects of spatial objects, it has higher accuracy compared to traditional methods and advantages in the process of modelling and prediction. GWR has been widely applied in exploring information and knowledge included in the night-time light images [59,60]. In this study, we will use GWR to model unbalanced development, which will be described in Section 2.3.6, by urban density which was defined in Section 2.3.2. Considering that GWR has systematic and complicated theories, technical details of how to use GWR refer to the literature [61].

2.3.6. Measuring Unbalanced Development in Urban Area

In economics, the Gini Index is widely used to measure inequality of income distribution, and higher Gini represents higher inequality of income inside a society [62,63]. Based on the same principle, to measure inequality of regional development, Elvidge et al. proposed the Night-time Light Development Index (NLDI), based on night-time light images and population density map [64]. The NLDI describes the inequality of night-time light per capita, by measuring the mismatch between night-time light and population, and it was applied for studies on unbalanced development in different areas [64,65]. Considering that the population density map includes a lot of uncertainties, and its accuracy is difficult to measure, we used the simplified NLDI, called the unbalanced index of night-time light (UINL), which only considers the distribution of night-time light inside urban area.

The principle of the UINL is simple: if all the night-time lights are concentrated in one spatial grid of urban area, the distribution of night-time light is totally unbalanced; contrarily, if all the lights are equally distributed in all the spatial grids inside a city, the distribution of night-time light is totally balanced. To illustrate the principle of constructing the UINL, a Lorenz curve, which was used to define the Gini Index, is drawn in Figure 4.

The steps for computing the UINL is as follows. Firstly, extract all the radiance values of the VIIRS images inside the urban mask at 500 m resolution, and sort the values from low to high. Consequently, a curve of cumulative percentages of night-time light, measured by radiance, is drawn as in Figure 4, and this is the Lorenz Curve. Finally, get the UINL by calculating the area size of the UINL = A/(A + B), as illustrated in Figure 4. Given that the night-time light is equally distributed in the urban land, A = 0, and UINL = 0; if night-time light is only concentrated in a small region of the urban land, UINL is very close to 1. These cases show that a larger UINL represents larger unbalanced development inside an urban region.

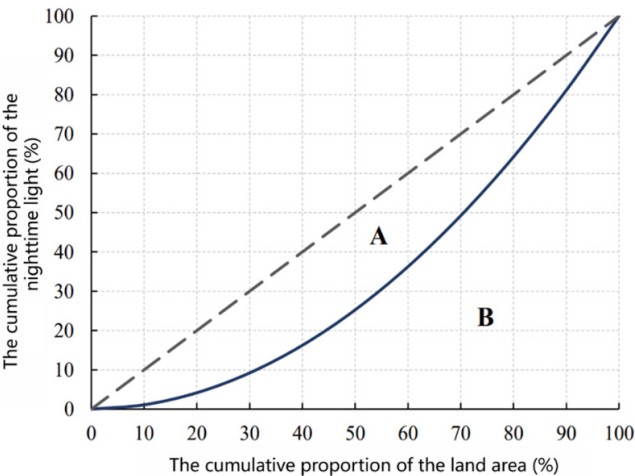

**Figure 4.** The Lorenz curve for calculating the Unbalanced Index of Night-time Light (UINL).

## 3. Results

### 3.1. Distribution and Evolution of Urban Size

Using the indexes of urban area and total night-time light, we calculate the urban sizes during 2012–2019 as shown in Figures 5 and 6. To keep the two sub-figures comparable in colors, the high values (also low values) were set to the same values. For example, in fact the largest urban area in 2012 is less than 4000 km$^2$, and it is larger than 4000 km$^2$ in 2019. This strategy of demonstration is also applied to Figure 6, Figure 11 and Figure 13. It was found that in 2012 the largest three cities in the YREB are Shanghai, Suzhou, and Xuzhou, measured in urban area, and the cities are Shanghai, Suzhou and Ningbo, measured in total night-time light. In 2019, the largest three cities in the YREB are Suzhou, Shanghai and Xuzhou, measured in urban area, while the three cities are Shanghai, Suzhou and Chengdu, measured in night-time light. We will analyze the urban sizes and distributions using rank-size law and transition matrix as follows.

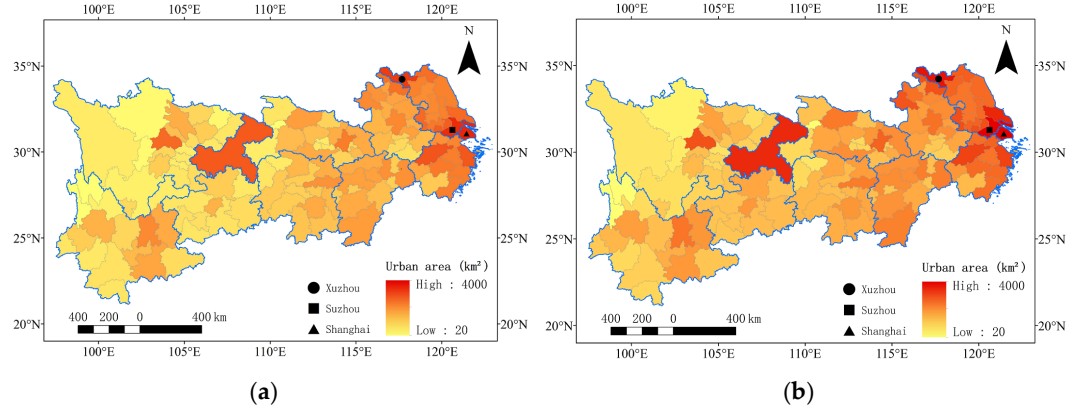

**Figure 5.** Urban area of 130 cities in YREB: (**a**) 2012; (**b**) 2019.

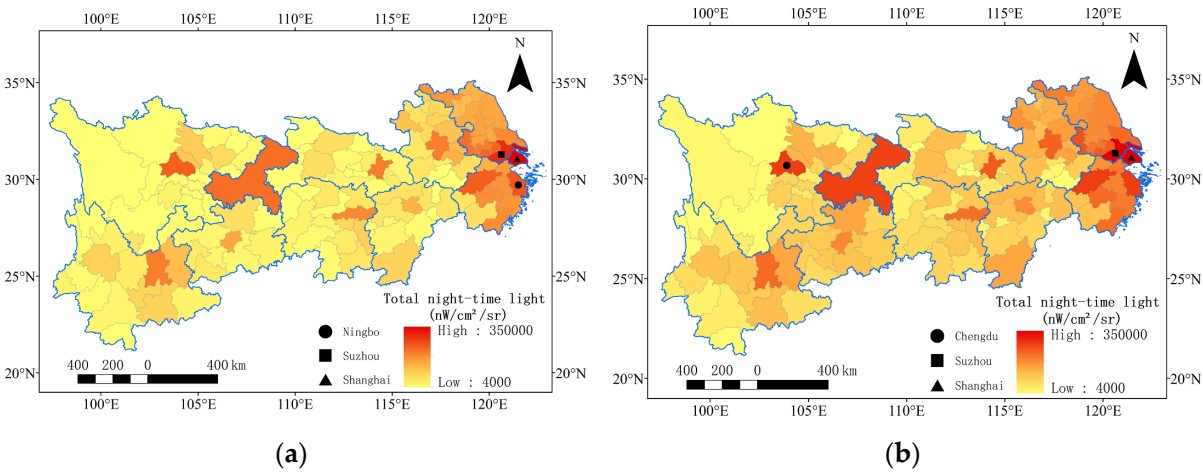

**Figure 6.** Total night-time light of 130 cities in YREB: (**a**) 2012; (**b**) 2019.

- Rank-size law

Using the rank-size law, we draw the rank-size curves based on urban area and total night-time light, with results shown in Figure 7. The rank-size model fits the data well, with $R^2$ values all larger than 0.6 for the four data groups as shown in Figure 7, suggesting that the rank-size model can reflect the urban size distribution. The $q$ values of the rank-size law based on urban area are 0.6039 and 0.5383 for 2012 and 2019, respectively. In contrast, those of the total night-time light are 0.9331 and 0.8187 for 2012 and 2019, respectively. We find that: (1) the $q$ values based on the night-time light are larger than those of the urban area, suggesting that the urban system of the YREB shows a higher aggregation degree when measured with night-time light compared to urban area; (2) the urban system in the YREB was becoming more dispersed during 2012–2019 as the $q$ values based on the two indexes are all reduced. In summary, the rank-size law analysis has different results when using urban area and total night-time light, but both indexes show the YREB is becoming more dispersed during 2012–2019.

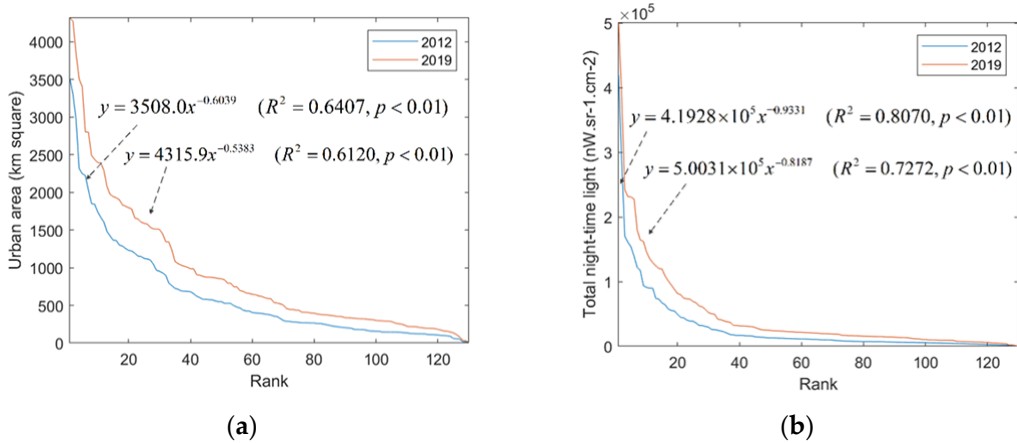

**Figure 7.** Rank-size curve of 130 cities in YREB: (**a**) Urban area; (**b**) Total night-time light.

- Transition probability

The above results show that the city ranks based on urban area and total night-time light are different, and both shift from 2012 to 2019. To quantify the shift pattern, we used the transition matrix, as introduced in Section 2.3.3. The results were listed in Tables 1 and 2. From the perspective of the urban area, cities in lower levels tend to transition to higher levels, with probabilities of 0.0328, 0.0714, and 0.1364, in transitions of A→B, B→C and C→D, respectively. For the highest ranked cities (e.g., types C and D), there is somewhat

of a downward trend in ranking, with probabilities of 0.2105 and 0.0909, in transitions of D→C and C→B, respectively. From the perspective of total night-time light, the shift in ranking is relatively larger; for example, type B has transitions to both types A and C, with probabilities of 0.1250 and 0.1875, respectively, while type B does not fall into type A measured in urban area. Similar findings exist for type C. In summary, transitions of urban sizes are different measured according to the two different indexes: (1) when measured with urban area, the city levels change less than that of the total night-time light; (2) cities with smaller sizes tend to rise in ranking based on urban area, but the change trend measured with night-time light is not obvious, as smaller sized cities can shift to both lower and higher levels with certain probabilities.

**Table 1.** The transition matrix for the evolution of urban size, measured with urban area, in YREB during 2012–2019.

| Original Class | Number of Transitions | | | | Probability of Transitions | | | |
|---|---|---|---|---|---|---|---|---|
| | **A** | **B** | **C** | **D** | **A** | **B** | **C** | **D** |
| A | 59 | 2 | 0 | 0 | 0.9672 | 0.0328 | 0 | 0 |
| B | 0 | 26 | 2 | 0 | 0 | 0.9286 | 0.0714 | 0 |
| C | 0 | 2 | 17 | 3 | 0 | 0.0909 | 0.7727 | 0.1364 |
| D | 0 | 0 | 4 | 15 | 0 | 0 | 0.2105 | 0.7895 |

**Table 2.** The transition matrix for the evolution of urban size, measured with total night-time light, in YREB during 2012–2019.

| Original Class | Number of Transitions | | | | Probability of Transitions | | | |
|---|---|---|---|---|---|---|---|---|
| | **A** | **B** | **C** | **D** | **A** | **B** | **C** | **D** |
| A | 74 | 10 | 0 | 0 | 0.8810 | 0.1190 | 0 | 0 |
| B | 2 | 11 | 3 | 0 | 0.1250 | 0.6875 | 0.1875 | 0 |
| C | 0 | 1 | 10 | 2 | 0 | 0.0769 | 0.7692 | 0.1538 |
| D | 0 | 0 | 1 | 16 | 0 | 0 | 0.0588 | 0.9412 |

*3.2. Urban Growth Patterns*

Figure 8 illustrates the urban growth rates based on urban area and night-time light in the YREB. From the perspective of growth, the three fastest growing cities are Tongren, Bazhong, and Diqing Tibetan Autonomous Prefecture, measured based on urban area, and the cities are Guangan, Qiannan Buyi, and Miao Autonomous Prefecture, and Zhangjiajie, measured with total night-time light. It is interesting to find that the Shennongjia Forestry District and Panzhihua are the regions with a decline in night-time light during 2012–2019. Panzhihua is a resource-depleted city [66], and Shennongjia Forestry District is a forestry-protected region where the population had moved out in recent years. Tables 3 and 4 list the stratified number of cities of different growth rates (*r*). It is clear that the cities in the Upper Reaches have the most rapid growth in urban area, with 44.68% of cities having $50\% < r \leq 100\%$ and 31.91% of them having $100\% < r$, while these rates are much lower in the Lower Reaches. In contrast, the Middle Reaches have the most rapid growth in night-time light, with 59.52% of cities having $100\% < r$, which is higher than the Lower Reaches and the Upper Reaches.

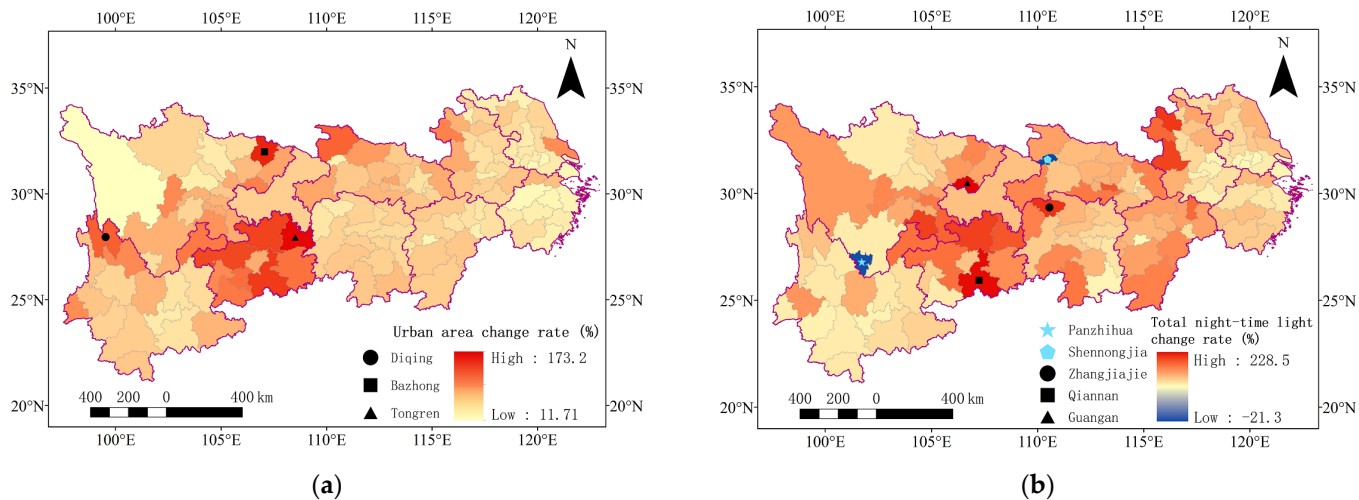

**Figure 8.** The urban growth rate during 2012–2019: (**a**) urban area; (**b**) total night-time light.

**Table 3.** Growth rates (*r*) of urban area of cities in the three reaches.

| Range | Cities in Upper Reaches | | Cities in Middle Reaches | | Cities in Lower Reaches | |
|---|---|---|---|---|---|---|
| | Number | Ratio (%) | Number | Ratio (%) | Number | Ratio (%) |
| $r \leq 25\%$ | 1 | 2.13 | 2 | 4.76 | 6 | 14.63 |
| $25\% < r \leq 50\%$ | 10 | 21.28 | 14 | 33.33 | 23 | 56.10 |
| $50\% < r \leq 100\%$ | 21 | 44.68 | 25 | 59.52 | 10 | 24.39 |
| $100\% < r$ | 15 | 31.91 | 1 | 2.38 | 2 | 4.88 |

**Table 4.** Growth rates (*r*) of night-time light of cities in the three reaches.

| Range | Cites in Upper Reaches | | Cities in Middle Reaches | | Cities in Lower Reaches | |
|---|---|---|---|---|---|---|
| | Number | Ratio (%) | Number | Ratio (%) | Number | Ratio (%) |
| $r \leq 25\%$ | 2 | 4.26 | 3 | 7.14 | 4 | 9.76 |
| $25\% < r \leq 50\%$ | 11 | 23.40 | 2 | 4.76 | 7 | 17.07 |
| $50\% < r \leq 100\%$ | 12 | 25.53 | 12 | 28.57 | 22 | 53.66 |
| $100\% < r$ | 22 | 46.81 | 25 | 59.52 | 8 | 19.51 |

As shown in Figure 9, the Upper Reaches have the fastest growth in urban area (e.g., 60.49%), and the Middle Reaches have the fastest growth in night-time light (91.11%), while the Lower Reaches have the lowest growth in both urban area (40.05%) and night-time light (51.66%).These results suggest that the Middle Reaches has a more compact growth than the Upper Reaches, although these two reaches both have rapid urban growth. Not surprisingly, the Lower Reaches have the lowest growth rates in both urban area and night-time light, but the absolute values of rates are still very high (40.05% and 51.66%).

Using the autocorrelation analysis, the LISA cluster map and significance map were generated, as shown in Figure 10. The values of global Moran' I index for change rates of urban area and night-time light are 0.3539 and 0.2617, respectively, indicating that the growth of urban area and night-time light tends to agglomerate in geography. From Figure 10, we learn that east of the Upper Reaches has the largest high-high agglomeration of urban area growth, while the Lower Reaches has the largest low-low agglomeration. For the night-time light growth, east of the Upper Reaches has the largest high-high agglomeration of high-high growth, while southwest of the Upper Reaches and center of the Lower Reaches have a major low-low agglomeration of night-time light growth.

These results suggest that the growth patterns of urban area and night-time light have clear spatial patterns, but they are different.

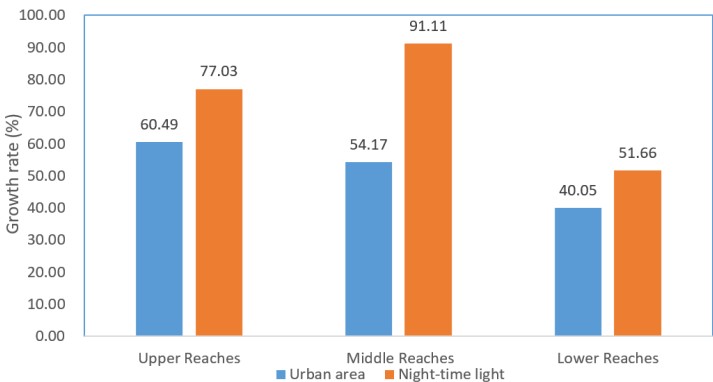

**Figure 9.** The urban growth rates of the three reaches in YREB.

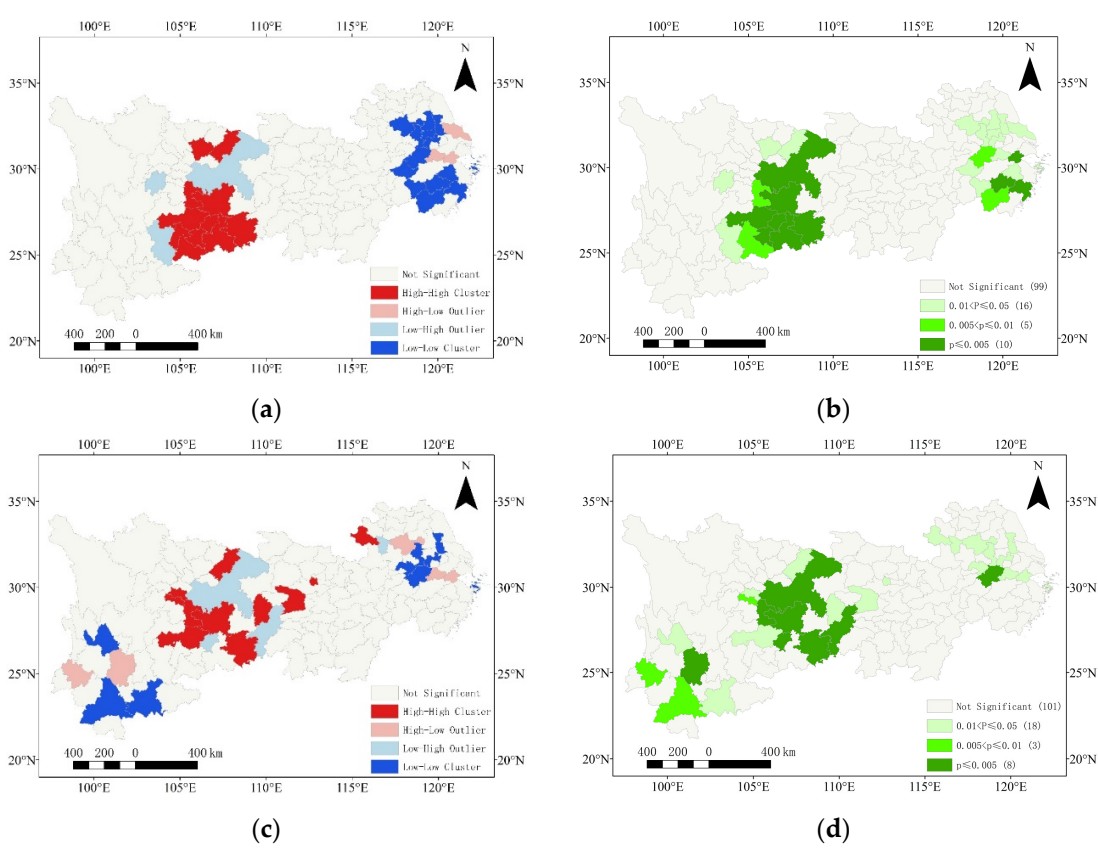

**Figure 10.** The autocorrelation analysis of urban growth: (**a**) LISA cluster map for urban area growth rate; (**b**) significance map for urban area growth rate; (**c**) LISA cluster map for night-time light rate; (**d**) significance map for night-time light growth rate.

### 3.3. Urban Density Patterns

Urban density, measured with the UNLD, was mapped over 130 cities in Figure 11, and the UNLD values for the three reaches were shown in Figure 12. In 2012, the three cities with the highest urban density are Shanghai, Panzhihua and Meishan, and the cities are Shanghai, Meishan and Nanchang in 2019. In addition, the three cities with the highest urban density growth are Yibin, Ezhou and Shangrao. It is also surprising to see overall urban density in Upper Reaches is the highest in both 2012 and 2019 (Figure 12a), this might be explained by the fact that cities in the Upper Reaches are more likely to be located in mountainous region where developable land is limited, so that the urban density has to

be high compared to the cities in the plain regions (e.g., the Middle Reaches and Lower Reaches). From Figure 12b, the three reaches all show that urban density increased during 2012–2019, while the Middle Reaches show the largest increase in urban density, and the Upper Reaches shows less of an increase in urban density but a still higher value than the Lower Reaches. The urban density pattern can explain why night-time light growth is much higher than urban area growth in the Middle Reaches, which was revealed in Section 3.2.

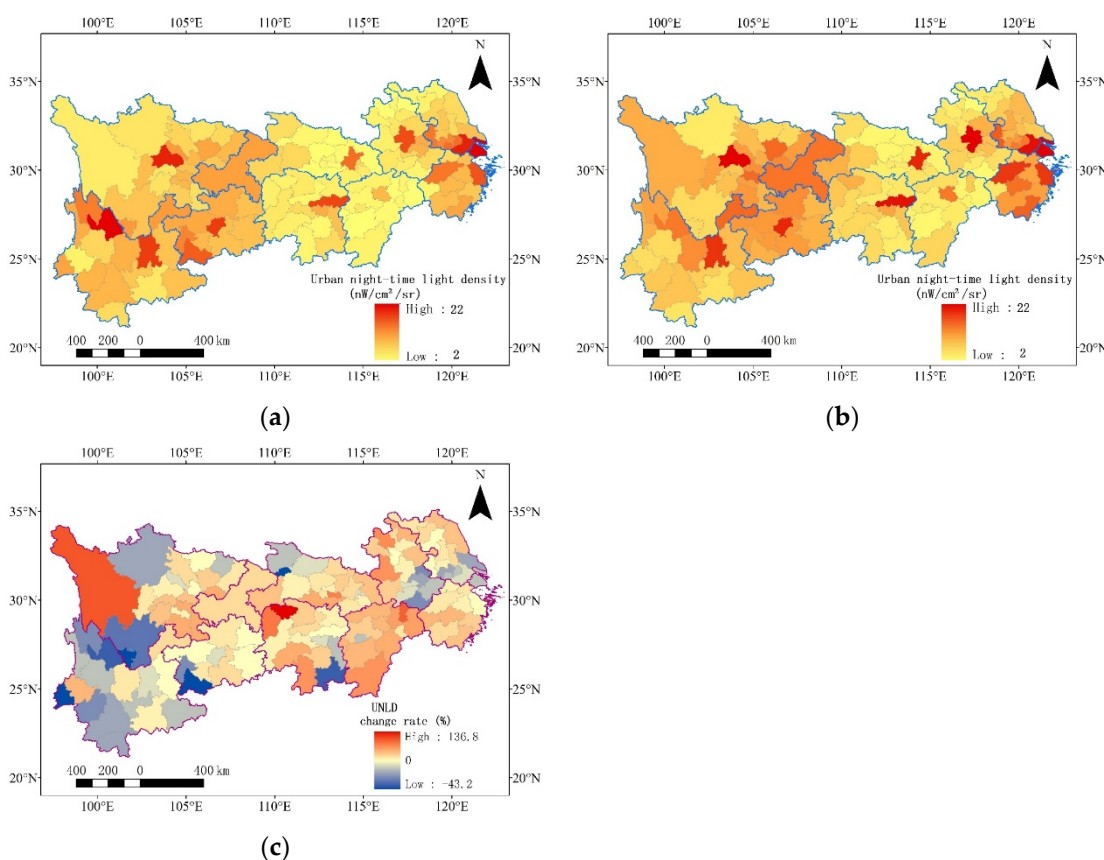

**Figure 11.** The urban night-time light density (UNLD) of YREB: (**a**) 2012; (**b**) 2019; (**c**) change rate.

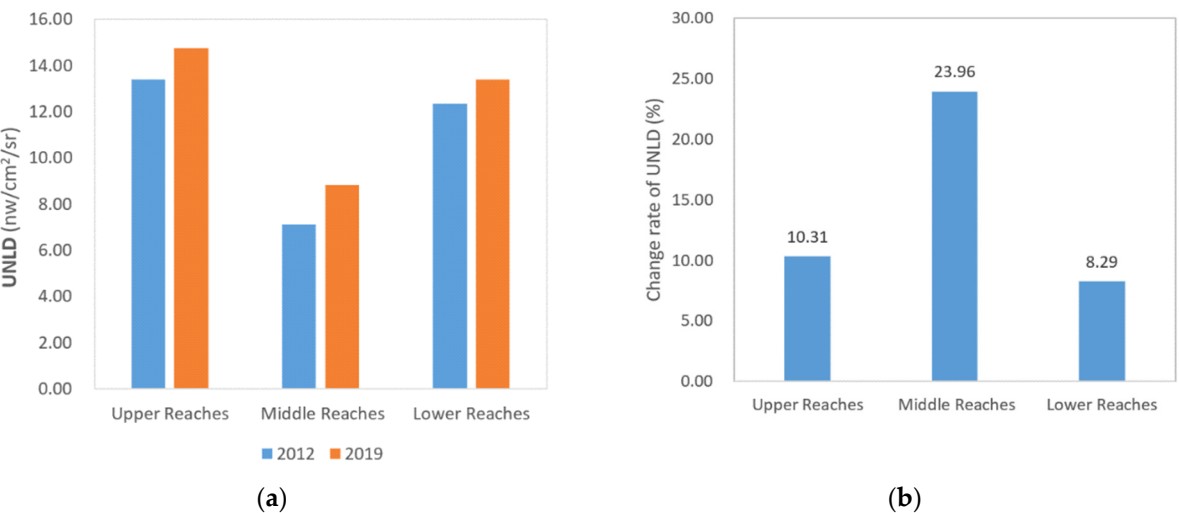

**Figure 12.** Urban night-time light density (UNLD) and its change rates: (**a**) UNLD for 2012 and 2019; (**b**) change rates.

To show the UNLD changes in different cities more clearly, we calculate the change rate of the UNLD (Figure 11c), and we define $r < -0.1$ as a decrease in UNLD, define $-0.1 \leq r \leq 0.1$ as stable, and $0.1 < r$ as an increase. In Table 5, we found that 21.28% of cities in the Upper Reaches have decreases in density, much more than do those in the Middle (4.76%) and Lower Reaches (4.88%), suggesting that the Upper Reaches have more urban sprawl cities, where urban efficiency declined, than in the Middle Reaches and Lower Reaches.

**Table 5.** Change rates (*r*) of urban density in 130 cities in the three reaches in YREB.

| Range | Upper Reaches | | Middle Reaches | | Lower Reaches | |
|---|---|---|---|---|---|---|
| | Number | Ratio (%) | Number | Ratio (%) | Number | Ratio (%) |
| $r < -0.1$ | 10 | 21.28 | 2 | 4.76 | 2 | 4.88 |
| $-0.1 \leq r \leq 0.1$ | 12 | 25.53 | 6 | 14.29 | 11 | 26.83 |
| $0.1 < r$ | 25 | 53.19 | 34 | 80.95 | 28 | 68.29 |

*3.4. Unbalanced Urban Development*

The above analysis focused on the growth in urban area and night-time light, showing the regional disparity among various cities and regions, while the intra-city disparity has not been evaluated. Using the UINL as defined in Section 2.3.6, we analyze the unbalanced development inside cities in the YREB. For all the 130 cities, the UINL for 2012 and 2019 was mapped. As shown in Figure 13, some spatial patterns are clear. Firstly, the west of the Upper Reaches is in a more unbalanced development pattern than that of the east in both 2012 and 2019; this finding is consistent with a previous analysis showing that regions with sparse populations develop in more unbalanced ways than do the densely population regions in China [67], and most of the cities in this region are becoming more unbalanced. Secondly, for the Middle Reaches, the north cities are obviously becoming more unbalanced, while several cities in the south become more balanced. Thirdly, the south of the Lower Reaches is more balanced than the north, and the south is becoming more balanced compared to the north. Although there are some disparities in the UINL for different cities inside reaches, the unbalanced development of different reaches as a whole do not show much difference: As shown in Figure 14, the UINL are all around 0.6 for the three reaches for both 2012 and 2019; the Upper Reaches are becoming more unbalanced, while the other two reaches are becoming more balanced although with very low change rates.

Previous studies suggest that regional inequality in Chinese cities is negatively correlated to urban density by analyzing DMSP/OLS night-time light [67]; therefore, we hypothesize that the unbalanced development is correlated with urban density in the YREB. To test this hypothesis, we make a linear regression chart between the UNLD and the UINL over 130 cities in the YREB. As Figure 15 shows, urban density is significantly negatively correlated with the UINL, with linear regressions $R^2$ of 0.2584 and 0.4258 for 2012 and 2019, respectively. As night-time light is an efficient proxy for infrastructure development [31], this analysis result shows that higher density of infrastructure and economic activities tends to correlate with more balanced development across cities, and this finding is consistent with previous studies using DMSP/OLS data [67].

To incorporate local spatial information for modelling, we model the UINL on the UNLD in the YREB based on the GWR, which was compared to the OLS regression (Figure 15). The following findings (Tables 6 and 7) were derived: (1) For the GWR method, its $R^2$ and $R^2$ adjustment values are both higher than 0.57, which shows much better performance than the OLS regression, indicating that the UINL can model the UNLD well, and that there is a positive relation between the two variables; (2) The corrected Akaike Information Criterion (AICc) is a measure of model performance; the model with a lower AICc value is considered to be a better model. In this study, the AICc value of GWR is significantly smaller than the AICc value of OLS, which means that the GWR,

considering the impact of geographical location, is more accurate in the regression between the UINL and the UNLD; (3) The residuals of a suitable regression model will be randomly distributed, and clustering of the residuals indicates that at least one key explanatory variable is missing. Calculating the global Moran's I index for the regression residuals, we can analyze whether the residuals are clustered. In this study, the Moran's I of the GWR residuals is less than 0.05 and not significant ($p > 0.1$), and the Moran's I of the OLS residuals is around 0.4 and significant ($p < 0.01$), indicating that the GWR model has a good regression performance, and it is better than the OLS model. The above findings suggest that higher urban density may generate more balanced development, and the relationship is stronger by considering local spatial information.

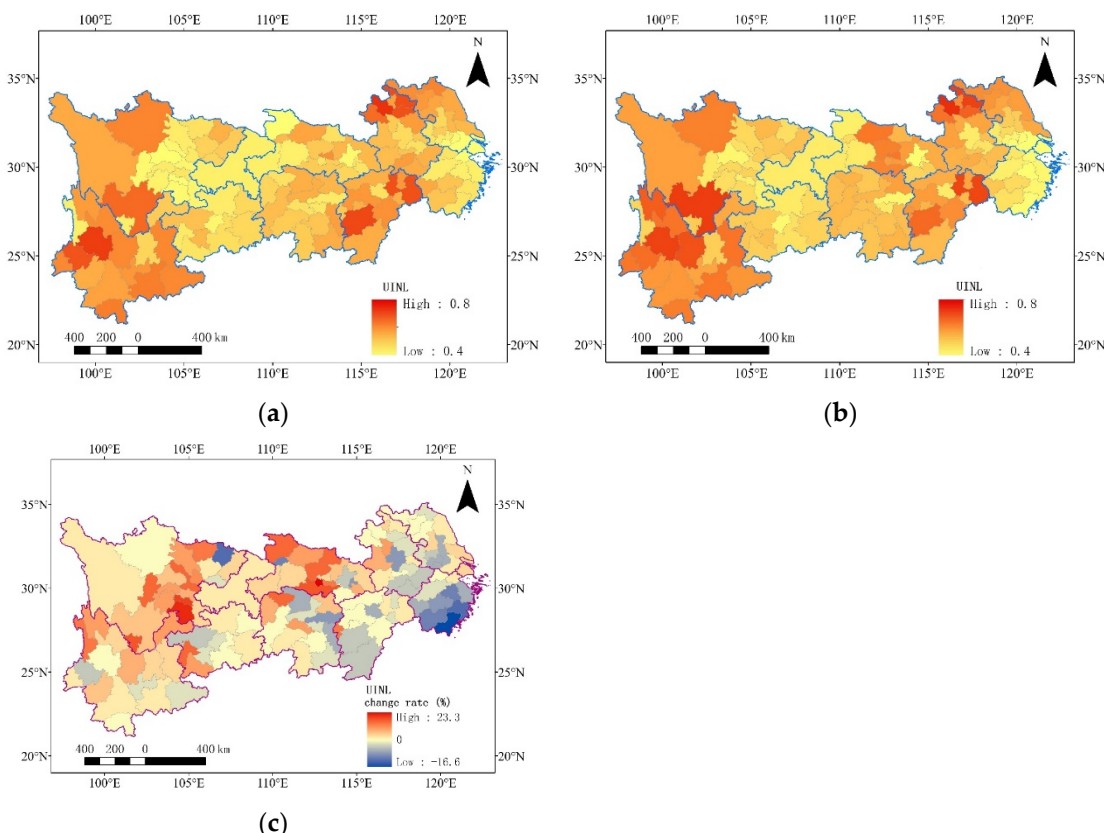

**Figure 13.** The UINL for the cities in the YREB: (**a**) 2012; (**b**) 2019; (**c**) the change rate.

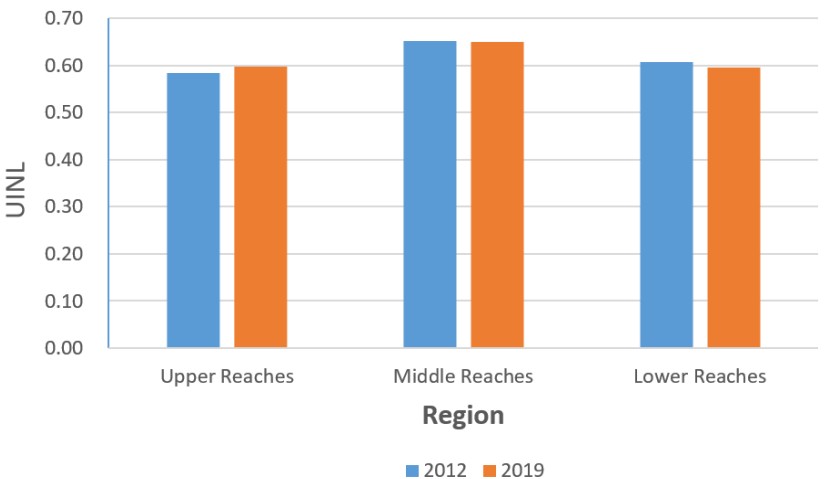

**Figure 14.** The UINL for three reaches in YREB.

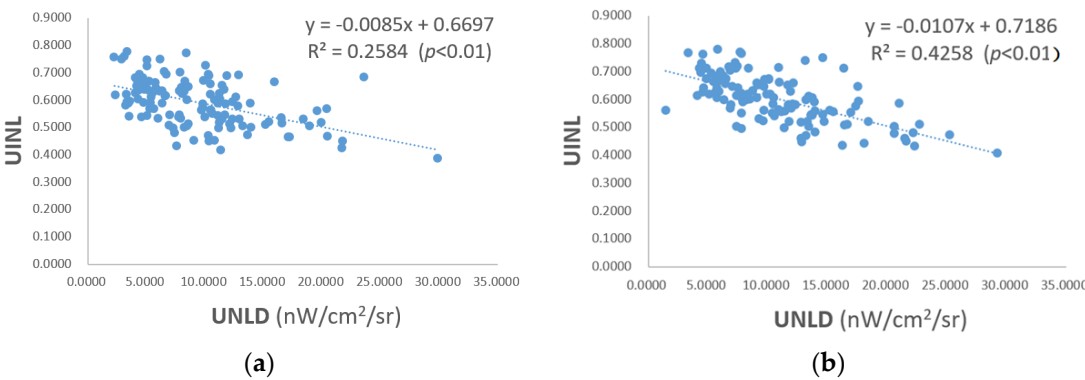

**Figure 15.** The OLS regression for UNLD and UINL: (**a**) 2012; (**b**) 2019.

**Table 6.** GWR and OLS regression results of UNLD and UINL for 2012.

| Variable | GWR | OLS |
|---|---|---|
| AICc | −368.604028 | −308.548948 |
| $R^2$ | 0.6570 | 0.2584 |
| $R^2$ Adjusted | 0.5763 | 0.2526 |
| Moran's I of regression residuals | −0.01465 | 0.4803 |

**Table 7.** GWR and OLS regression results of UNLD and UINL for 2019.

| Variable | GWR | OLS |
|---|---|---|
| AICc | −386.039664 | −341.888953 |
| $R^2$ | 0.6982 | 0.4258 |
| $R^2$ Adjusted | 0.6278 | 0.4213 |
| Moran's I of regression residuals | 0.02668 | 0.39615 |

## 4. Discussion

### 4.1. Contribution and Findings

Optimizing regional development in the YREB is a national strategy of the Chinese central government, and there have been a number of discussions on the urban issues in the YREB [17,28,29]. Urban size is viewed as an important indicator for regional development processes, and urban population [49,68] and urban area [69,70] are two commonly used measures for urban size.

Considering that the population census in China is taken only every 10 years, the availability of urban population data is limited. In comparison, urban area can be estimated from satellite imagery from both night-time light imagery [71] and daytime imagery [72], and thus it has been used for analyzing the urban size distribution and evolution in YREB. The night-time light imagery was mainly used to extract the urban boundary in YREB [26,27,71,73]. However, due to coarse resolution of night-time light images and uncertainties about applying the threshold [74], the extracted urban boundaries involve a lot of uncertainty; therefore, researchers have turned to using the Landsat imagery to extract urban area data in the YREB [72]. In fact, there are a number of remote sensing based land cover products at medium resolution [44,75,76], which provide more accurate urban boundary information than that from the night-time light products. In this study, to analyze the urban size distribution and evolution, we made use of the urban land cover product, and this strategy would help to get more accurate information on the urban areas.

While previous studies mainly employed urban area as the index for urban analysis in the YREB, this study analyzed the urban size from two perspectives, urban area and

night-time light. The night-time light is able to reflect development of urban infrastructure as well as economic activities, so it has been widely used as an economic proxy by economists [30,36], as well as for analyzing urban size and evolution by geographers [77,78]. Combining the two indexes, the evolution of urban system in the YREB can be revealed from different perspectives. Notably, the urban night-time light density (UNLD), a newly derived index, has been analyzed for the YREB based on the two variables, and the evolution of this index helps to understand different expansion patterns. For example, we found that the Upper Reaches has more urban sprawl than the Middle Reaches and Lower Reaches, a pattern that is consistent with previous findings that the urban sprawl in the YREB is uneven [15]. In addition, cities in the YREB have increases in urban density, which is viewed as proxy for urban efficiency, as shown in Table 5 and Figure 14. This finding supports the previous study that urban efficiency of the YREB is increasing according the trend shown [17]

We have to know that urban density, measured with UNLD, is purely retrieved from remote sensing products, the GISA dataset, and VIIRS night-time light imagery. Considering that the remote sensing data can be acquired objectively compared to statistical data, this study provides a new and simple way to analyze urban evolution patterns, which helps to better understand the urban evolution in the YREB. Moreover, considering that the proposal method is totally based on a remote sensing product that is available to the public, it would help central and local governments in China to monitor the urban sprawl in the YREB with less cost in budget and time.

*4.2. Limitation and Future Work*

In this study, we made analyses for 2012 and 2019. The limited temporal information hindered us from comprehensively understanding the urban evolution in the YREB. The reason for using limited temporal data was that both the GISA and the VNP46A2 Black Marble have some annual fluctuation from data abnormality; therefore, we only compared and analyzed the data for 2012 and 2019, of which the temporal duration was long enough so that the data abnormality did not affect the result significantly. However, future studies should make full use of urban extent and night-time light information in the long-term, so that urbanization patterns in the YREB can be reconstructed. We acknowledge that this planned future work depends on urban land cover products and VIIRS night-time light with improved quality [42,43]. From a general perspective, night-time light images have suffered from a blooming effect that makes the dark area around the urban region appear bright due to the skyglow phenomenon [79]. Since no existing night-time light products have corrected it, this issue may have introduced some uncertainty for the analysis result in this study.

Actually, higher resolution of urban mask data helps to estimate the indexes such as the UINL and the UNLD more accurately. The spatial resolution of the VIIRS night-time light imagery is 740 m, which was resampled to around 500 m in the Black Marble product, while the urban land cover data from the GISA product has a 30 m resolution; this indicates that a large amount of land cover information was discarded as it was converted to 500 m resolution to match the night-time light imagery. For example, the UINL and UNLD are calculated based on 500 m resolution data, and thus these two variables were not estimated accurately. In addition, we directly used the definition of urban area that considers the impervious proportion to be more than 20% larger than in previous studies [45]. However, to what extent this kind of urban map can differentiate an urban area from a rural settlement still needs careful investigation for the YREB in future work. In 2021, the Chinese Academy of Sciences (CAS) launched the SDGSAT-1 satellite, which is able to map night-time light at 10 m resolution, and the data has global coverage with a number of applications already [80,81]. Future work will take such fine resolution images to match the urban land cover product at 30 m resolution, and that will help to retrieve urban information in the YREB more accurately.

This study follows research methods that made use of satellite-observed night-time light combined with other data to analyze urban growth without considering details of land use change [35,52,82–85]. However, the land use change information, which was not used in this study, is crucial for understanding the aftermath of urbanization. For example, it is important to know whether the new urbanized area was converted from previous agricultural land or not, as protecting agricultural land is a state policy for food security in China. Similarly, other natural land uses such as forest, grassland, and water are also important for environmental protection. Thus, in future studies, it would be valuable to utilize more information on past land use and changes in it to systematically evaluate whether economic and environmental development are balanced against the background of urbanization in the YREB.

This study focused on the spatiotemporal patterns of urban development in the YREB. The factors behind these patterns, which are important to understanding urbanization in the YREB, have not been discussed. For example, previous studies show that the GDP, total fixed asset investments, and urban population affect urban expansion in the YREB [40], and economic linkage can reduce the speed of urban land expansion in the YREB [86]. Thus, it is also important to apply policy analysis such as quantifying policies of urban planning, industry and environmental protection as well as their impacts, which are important to understand urban development in the YREB, by using attribution analysis methods.

## 5. Conclusions

This study proposed the use of two remote sensing based indexes, urban area and night-time light, to characterize the urban development patterns in the YREB during 2012–2019. Findings show an overall increase in urban density and efficiency but great regional disparity in urban development inside the YREB. Firstly, the urban system is evolved to be more dispersed according to both of the indexes, and the urban system is measured to transition less when gauged according to urban area rather than according to total night-time light. Secondly, the Upper Reaches has the largest growth rate of urban areas, the Middle Reaches has the largest growth rate in terms of night-time light, and the Lower Reaches has a relatively low growth rates using either of the two indexes. Thirdly, although most of the cities in the YREB have increased in urban density, the Upper Reaches has experienced more urban sprawl than the Middle Reaches and Lower Reaches; the Middle Reaches has shown more compact growth. Finally, the unbalanced development inside different reaches are very similar, and higher urban density and efficiency tends to be related to more balanced development.

This study suggests that combining two widely used indexes of urban size helps to provide more insight into urban development in the YREB, and this methodology can be combined with existing methods of urban efficiency evaluation. It can also be expanded for use on a national or global scale.

**Author Contributions:** Conceptualization, H.X., S.H. and X.L.; methodology, H.X. and S.H.; software, H.X. and X.L.; data curation, H.X. and S.H.; project administration, H.X.; writing, H.X. and X.L.; funding acquisition, H.X. and S.H.; supervision, S.H. All authors have read and agreed to the published version of the manuscript.

**Funding:** This study is supported by Hubei Provincial Natural Science Foundation of China under grant No. 2021CFB172, and National Natural Science Foundation of China under grant No. 42171272 and 42271371.

**Data Availability Statement:** The Black Marble product was downloaded from NASA, webpage https://ladsweb.modaps.eosdis.nasa.gov/missions-and-measurements/products/VNP46A4/ (accessed on 27 December 2022); GISA dataset was downloaded from Zenodo, webpage https://zenodo.org/record/6476661#.Y1Sh9bZBw2y (accessed on 27 December 2022).

**Conflicts of Interest:** The authors declare no conflict of interest.

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
