# Peer review of "Urban Distribution and Evolution of the Yangtze River Economic Belt from the Perspectives of Urban Area and Night-Time Light"

_land, doi:10.3390/land12020321_

Round 1
Reviewer 1 Report
This paper has two contributions. Firstly, it combined night-time light and urban area to assess the urbanization patterns while the traditional way of using night-time light mainly takes it to delineate the urban extent. Secondly, the proposed methods were carried out for YREB, which is a focus issue of urbanization in China. However, I have a few questions for the authors.
1. How to generate the Lorenz curve has not been described, please add more details on it.
2. How does cloud impact the night-time light image quality so as to impact the analysis result? I suggest the paper should discuss this issue.
3. The VNP46A4 has several layers such as all-angle products, the paper should describe which layer has been utilized.
4. In Figure 12(a), there are two back brackets. The authors should check over the entire paper to avoid similar problems.
5. Although the paper has found some patterns of urbanization in YREB, the implication of the findings have not been discussed which would improve the value of this paper.
6. For the first time, the abbreviation such as NASA should be used after the full term.
Author Response
Dear Reviewer:
Thank you for reviewing this manuscript.
I have a attached a letter to address the issues from the comments.
Huimin Xu

Reviewer 2 Report
The study explored the urban expansion in the Yangtze River Economic Belt region of China by using remote sensing land use and night light data and various spatial analysis methods. The data sources and methodology are clearly presented. The results are demonstrated well. It is recommended to publish after the following minor revisions:
1) please add province names to figure 1 so to help international readers follow the narrative
2) there are several cities and regions are highlighted in the results section, it is suggested to label them on the result maps
3) the authors need to discuss the drawbacks of the data in the discussion section. Is night light a good tool, is there any odd that night light does not reflect human activity? the paper used Landsat data to generate urban land use, could the Landsat data separate urban and rural residential land well?
Author Response

(The authors gave the same response as above.)

Reviewer 3 Report
The study focused on the changing pattern of urban distribution based on the urban area and night-time light density in YREB of China. The strategy used in this study may be a good way to figure out how cities grow by looking at proxy variables, like nighttime light intensity. Even though the authors did a good job of organizing the whole manuscript, there are still some things to think about to make the manuscript better as a whole.
Comments and suggestions:
1. There are several ways to figure out how urban growth is distributed. One of the most common ways to do this is to look at how land use and land cover have changed over time by looking at historical satellite images. Understanding how LULC patterns change over time gives the researcher insight into several important things, such as the process of urbanization, i.e., how natural, agricultural, and other types of lands are turning into cities and what that might mean for food and environmental security in the future; the factors that cause LULC changes; and the policies that are needed to make sure that economic and environmental development are balanced. Without taking into account the elements that impact the urbanization process, it is still questionable how the suggested research might be robust enough to be employed as a replacement to the prior techniques of studying urban sprawl and distribution.
2. Most studies done in China use a map of the whole country, including the South China Sea. It's strange that the author only uses the map of YREB. I think authors may put up a map of the entire country and point out the YREB as the case study area.
3. The authors included a spatial autocorrelation map (LISA cluster map) without Moran's I index and significance map. The author could include Moran's I index and significance map along with the LISA cluster maps.
4. Using data with a resolution of 500m to measure light intensity may make the reader doubt the accuracy of the assessment. Is it possible to improve the accuracy by analyzing 30m or 10m resolution satellite images? Perhaps the measurement's precision hasn't been mentioned by the authors. But it's important to know this!
5. The Markov Chain model is used, but nothing said about the possible uncertainty. Since uncertainty is linked in modeling, how did the author deal with the uncertainties in the study results?
6. The authors should meticulously check for typos and grammatical errors, especially between lines 256 and 264.
Author Response

(The authors gave the same response as above.)

Round 2
Reviewer 3 Report
The authors made improvements to their manuscript in response to the reviewers' comments, which eventually improved the overall quality of the contents. I have no more concerns about it and feel that it should be accepted for publication.
Author Response
Thank you so much!